# On the Use of Pulsed UV or Visible Light Activated Gas Sensing of Reducing and Oxidising Species with WO_3_ and WS_2_ Nanomaterials

**DOI:** 10.3390/s21113736

**Published:** 2021-05-27

**Authors:** Ernesto González, Juan Casanova-Chafer, Aanchal Alagh, Alfonso Romero, Xavier Vilanova, Selene Acosta, Damien Cossement, Carla Bittencourt, Eduard Llobet

**Affiliations:** 1Electronic Engineering, Uiversitat Rovira i Virgili, 43007 Tarragona, Spain; ernesto.gonzalez@urv.cat (E.G.); juan.casanova@urv.cat (J.C.-C.); alagh.aanchal@urv.cat (A.A.); alfonso.romero@urv.cat (A.R.); eduard.llobet@urv.cat (E.L.); 2Chimie des Interactions Plasma e Surface (ChIPS), Research Institute for Materials Science and Engineering, Université de Mons, 7000 Mons, Belgium; selene.acostamorales@umons.ac.be (S.A.); carla.bittencourt@umons.ac.be (C.B.); 3Materia Nova, Parc Initialis, 7000 Mons, Belgium; damien.cossement@materianova.be

**Keywords:** gas sensing, pulsed light modulation, FFT, PCA, PCR, NO_2_, NH_3_

## Abstract

This paper presents a methodology to quantify oxidizing and reducing gases using n-type and p-type chemiresistive sensors, respectively. Low temperature sensor heating with pulsed UV or visible light modulation is used together with the application of the fast Fourier transform (FFT) to extract sensor response features. These features are further processed via principal component analysis (PCA) and principal component regression (PCR) for achieving gas discrimination and building concentration prediction models with R^2^ values up to 98% and RMSE values as low as 5% for the total gas concentration range studied. UV and visible light were used to study the influence of the light wavelength in the prediction model performance. We demonstrate that n-type and p-type sensors need to be used together for achieving good quantification of oxidizing and reducing species, respectively, since the semiconductor type defines the prediction model’s effectiveness towards an oxidizing or reducing gas. The presented method reduces considerably the total time needed to quantify the gas concentration compared with the results obtained in a previous work. The use of visible light LEDs for performing pulsed light modulation enhances system performance and considerably reduces cost in comparison to previously reported UV light-based approaches.

## 1. Introduction

Over the past few decades, many research efforts have been directed towards indoor and outdoor air quality monitoring. The direct relation between environmental pollutants and human health has promoted the research on this topic. According to the World Health Organization, about 7 million people die every year caused by diseases related to air pollution [1]. Exposure to gases present in the atmosphere due to industrial activity, such as NH_3_ and NO_2_ can cause skin and eye damage and affect the respiratory and cardiovascular systems [2,3,4].

Some different operating principles such as electrochemical [5,6], optical [7,8], or chemiresistive have been used for gas sensing [9,10,11,12] related to air quality monitoring. One of the most studied approaches has been the use of metal oxides (MOX) chemiresistors due to their high sensitivity and the relatively simple associated driving and readout electronics, which confers them enormous versatility for being employed in a wide range of different applications, such as toxic and combustible gas detection, biosensing, environmental safety, and food quality control [13,14,15,16,17,18,19,20]. The operating principle of MOX sensors relies on surface redox reactions. Target gas molecules interact with oxygen species trapped at the sensor surface, thus, changing the electronic charge distribution in the sensing material, which eventually results in a resistance change [21,22,23,24].

Typically, MOX sensors have been operated highly above room temperature, at a few hundred degrees centigrade to enable surface reactions and achieve high sensitivity and baseline recovery. Heating supposes an important power consumption issue, especially for non-MEMS sensors, making them not suitable for portable or low-power applications [25,26,27]. Nevertheless, some different techniques have been employed during the last years to solve the problem generated by power consumption issues. The use of thermal modulation, UV-light irradiation at room temperature, and UV light activation combined with mild temperature heating, instead of working with just thermal activation at high temperatures have gained prominence [16,26,28,29,30,31,32,33]. The photoconductivity effect caused by the UV light irradiation creates electron-hole pairs, which increase the density of carrier charges along the semiconductor, making an acceleration of the absorption/desorption mechanism [34]. The use of UV light irradiation not only makes the sensor response of MOXs higher at low or even at room temperature but also shortens the time needed to reach the steady-state and to recover the sensor baseline [34,35]. Light enhanced gas sensing has been also applied on perovskite and metal transition dichalcogenides [36,37,38,39]. Although the light activation (constant light irradiation throughout all the measurement time or during the baseline recovery time) has been widely used for enhancing the sensing performance using light sources with a wavelength from the UV to the visible spectrum, only very few works present the study of a pulsed light mechanism. The use of UV light modulation with MOXs through a pulsed light activation mechanism has been employed to quantify gas concentration. This method is carried out by using the resistance changes induced by the pulsed light, which creates a ripple on the sensor resistance curve [31,40,41]. The information extracted from the resistance transients is used to establish a relationship with the target gas concentration. In addition, this method shortens response time and the humidity effect on sensing performance is reduced as well [40,41]. This methodology was also employed in the development of a portable system for the detection of NO_2_ at ppb levels [42].

On the other hand, some researchers have studied the quantification of target gas concentration and different gas discrimination using mathematical and statistical methods. Multivariate methods such as PCA, PCR, and machine learning have been employed for this purpose [43,44,45,46,47]. The combined use of electronic noses that employ arrays of sensors with the aforementioned methods has been applied to discriminate and quantify gases (e.g., NO_2_, ammonia, ethanol, acetone) [43,47,48,49,50,51,52,53]. Most of these works implement the mentioned data analysis by using sensor response feature vectors as input for the multivariate and machine learning approaches. Nevertheless, a few researchers have reported the use of the fast Fourier transform (FFT) components obtained from the sensor resistance transient as inputs for the data analysis strategies [54,55]. Employing this last approach, we have developed a methodology for quantifying NO_2_ using UV light modulation and FFT analysis of the sensor response signals from n-type metal oxide sensors [56]. However, pulsed UV light, n-type metal oxide sensors were found to lack accuracy at quantifying reducing species such as ammonia.

In this paper, we refine further and generalize our approach for quantifying both oxidizing and reducing species using light-pulsed chemiresistive sensors. For generalizing the methods, n-type (WO_3_ and SrTiO_3_@WO_3_) and p-type (WS_2_) sensors were synthesized and measured under combined low temperature and pulsed UV or visible light modulation. The inclusion of a p-type chemiresistor enabled the reliable quantification of reducing species, which had not been achieved before. The development of PCR models and their validation process for quantifying NO_2_ and NH_3_ concentration using FFT components from the analysis of the response transients is discussed. PCA is used to identify when sensors are exposed to NO_2_ or NH_3_. The refinements implemented enable reducing the time needed to successfully quantify the target gas concentration and improve model accuracy at estimating gas concentrations. These new findings expand the opportunities of using pulsed light chemisensing in different real applications.

## 2. Experimental Set-Up

### 2.1. Sensor Fabrication

#### 2.1.1. Strontium Titanate Loaded Tungsten Trioxide Sensors

Tungsten trioxide nanoneedles (NNs) functionalized with strontium titanate nanoparticles were grown using a one-step process of aerosol assisted chemical vapor deposition (AACVD) which is a widely used technique for synthesizing MOX nano and microstructures [57]. Materials were grown on top of a commercial alumina substrate from Ceram Tech GmBH, with screen-printed, interdigitated platinum electrodes (300 μm gap) on the front side and an 8 Ω screen-printed heater on the backside. In a mixture of 24 mL of acetone (CAS: 67-64-1) and 9 mL of methanol (CAS: 67-56-1), 50 mg of tungsten hexacarbonyl (W(CO)_6_) (purity 97%, CAS: 14040-11-0) were dissolved. Following this, 5 mg of strontium titanate nanopowder (CAS: 12060-59-2) were dispersed inside the solution using an ultrasonic bath. Nitrogen (N_2_) was used as a carrier gas to transport the aerosols generated by means of an ultrasonic humidifier at a flow of about 800 sccm. The total transport of the aerosols and the deposition process took about 40 min. The deposition chamber temperature was kept at 400 °C during all the deposition processes and then naturally cooled down to room temperature. After the one-step growth in the AACVD, an annealing process was performed at 500 °C for 2 h in a Carbolite CWF 1200 muffle furnace, to fully oxidize the WO_3_ and remove the residual carbon from the precursor.

#### 2.1.2. Tungsten Trioxide (WO_3_)

Pure WO_3_ sensors were fabricated using the same procedure and equipment described for the case of SrTiO_3_@WO_3_ sensors, but without including the strontium titanate nanopowders. In this case, 50 mg of tungsten hexacarbonyl were dissolved in a mixture of 15 mL of acetone and 5 mL of methanol, and the rest of the conditions were kept equal to those in the previously described synthesis.

#### 2.1.3. Tungsten Disulphide (WS_2_)

Multi-layered nanosheets of WS_2_ were synthesized in two steps. First, WO_3_ NNs were grown using AACVD as described above. During the second step of synthesis, the as-grown WO_3_ nanomaterial was sulfurized to form WS_2_ in a quartz tube furnace using an atmospheric pressure chemical vapor deposition technique (CVD) under hydrogen-free conditions. Before the sulfurization process, the quartz tube was flushed with 0.5 L/min of argon gas to remove any oxygen present in the reactor. Two ceramic boats containing an equal amount of sulfur (S) powder (>99.95%, Sigma Aldrich, CAS: 7704-34-9) were placed at different temperature zones of the deposition furnace. Furthermore, a smaller semi-sealed quartz tube loaded with substrate containing nanoneedles of WO_3_ with a boat carrying S precursor was introduced inside the larger quartz tube, such that both the substrate and the S boat are positioned at the center of the deposition furnace. Afterward, a second boat carrying an equal amount of S powder was introduced inside the upstream of the bigger quartz tube. Then the furnace was heated from room temperature to 900 °C with a heating rate of 40 °C/min to remove the contaminants, such as water or residual organics to obtain the nucleation of WS_2_. The growth of WS_2_ was kept at 900 °C for 30 min under a constant flow of argon. After the growth phase, the furnace was cooled naturally to room temperature.

### 2.2. Morphological Characterization Systems

The different sensors were characterized via Field Emission Scanning Electron Microscope (FESEM), Energy-dispersive X-ray Spectroscopy (EDX), Raman Spectroscopy, X-ray Diffraction (XRD), X-ray Photoelectron Spectroscopy (XPS), and Time-of-Flight Secondary Ions Mass Spectrometry (ToF-SIMS). The FESEM–FIB Scios 2 from FEI Company was used to obtain images from the sensor surface to analyze nanostructure growth and distribution. Sample characterization was performed at high-vacuum, and the electron acceleration voltage was established between 2 and 5 kV. EDX incorporated in the FESEM–FIB Scios 2 was used to check the chemical composition of the sensors.

An FT-IR Raman spectrometer from Renishaw and the DM2500 confocal microscope from Leica Microsystems were used to perform the Raman spectroscopy analysis. Laser sources with a wavelength of 514, 633, and 785 nm were used. The laser beam power was set at 0.1%.

XRD measurements were made using a Siemens D5000 diffractometer (Bragg–Brentano parafocusing geometry and vertical θ-θ goniometer) fitted with a curved graphite diffracted-beam monochromator, incident and diffracted -beam Soller slits, a 0.06° receiving slit, and scintillation counter as a detector. The angular 2θ diffraction range was between 5 and 70°. The data were collected with an angular step of 0.05° at 3 s per step and sample rotation. Cukα radiation was obtained from a copper X-ray tube operated at 40 kV and 30 mA.

For XPS experiments a VERSAPROBE PHI5000 spectrometer from Physical Electronics, equipped with a monochromatic AlK X-Ray was used. The energy resolution was 0.6 eV. A dual beam charge neutralization composed of an electron gun (~1 eV) and the Argon Ion gun (≤10 eV) was used for compensation of charge built up on the sample surface during the measurements. All binding energies were calibrated to the C 1s peak at 284.6 eV. The CASA XPS software was used for spectra analysis.

The ToF-SIMS experiments were conducted on a TOF-SIMS IV instrument from ION-TOF GmbH (Münster, Germany). Prior to the analysis, the sample surface (600×600 µm) was sputter cleaned using O_2_ ions accelerated at 1 kV, for 120 s. For recording the m/z spectra, a pulsed 25 kV Bi1+ ion beam rastered during 300 s over an area of 100×100 µm^2^, was used. The total ion fluence was kept under 1012 ions per cm^2^ in order to assure static conditions. The secondary ions were extracted at a 2 kV acceleration voltage. Positive spectra were calibrated to the H+, C+, CH+, CH_2_+, CH_3_+, C_2_H_3_+, and C_2_H_5_+ peaks.

### 2.3. Gas Measuring System Description

Measurements were made inside a Teflon chamber with an inner volume of about 21 cm^3^. The chamber is totally isolated from the ambient light and has the capacity to hold up to four sensors at the same time, which allows the use of the three types of sensors synthesized at the same time. LEDs are inserted in the chamber top through two air-tight connection joints, staying at about 7.5 mm from the sensor surface, which allows homogeneous irradiation. Connectors in the back side of the chamber allow control of the sensors operating temperature and measure the resistance of the sensing layers. Sensor resistance is measured and recorded every 1 s by using a Keysight 34972A LXI Data Acquisition/Switch Unit controlled with BenchLink Data Logger 3 from Agilent Technologies.

Gas concentrations established to test the sensors were set by mean of a mass-flow controller system (EL-FLOW®) from Bronkhorst, using Flow View and Flow Plot software from the same company. NO_2_ and NH_3_ flows coming from calibrated cylinders with 1 ppb and 100 ppm respectively (balanced in synthetic air) were mixed in adequate proportions with a synthetic air flow coming from a zero-grade air cylinder. The total flow across the chamber was kept at 100 mL/min during all the measurements.

To build and validate the models presented in this work, sensors were exposed to NO_2_ concentrations of 250, 500, and 750 ppb, and NH_3_ concentrations of 25, 50, and 75 ppm. Gas concentrations were selected to be under the exposure limits established by the Occupational Safety and Health Administration (OSHA) permissible exposure limit (PEL), and the National Institute for Occupational Safety and Health (NIOSH) recommended exposure limit (REL), from the United States. Every gas cycle made was composed of 3 different NO_2_ or NH_3_ concentrations, using 15 min of gas exposure and then 1 h of baseline recovery under synthetic air. Appendix A shows a schema of the gas measurement system used.

### 2.4. Light Pulse Generation

UV and visible light modulations were carried out using LEDs with an emission wavelength of 365 (MT3650W3-UV from Marktech Optoelectronics) nm and 410 nm (OSV5HA5A32A from Optosupply), respectively. An electronic circuit was designed and implemented to control the forward current of the LEDs. To power up the control circuit and set the activation and deactivation periods of the LEDs, an Arduino Mega 2560 from Arduino was used. Digital outputs and timers from the Arduino were used for this purpose.

### 2.5. Data Analysis Process Description

In order to generate a quick pathway to quantify both oxidizing and reducing gases concentration some mathematical and computational tools such as fast Fourier transform (FFT), PCA, and PCR have been used. The data analysis process implemented to quantify the gas concentrations was carried out using Matlab R2020a. 

In contrast to traditional methods to characterize chemiresistive gas sensors and quantify gas concentrations, where the steady-state response of the sensor resistance and then the full baseline recovery is needed, the methodology presented in this work requires just a few minutes to accurately quantify the studied gas concentration. Figure 1 depicts the flow diagram from the data analysis process.

Similar to the methodology presented in [56], a frequency domain analysis is performed to the sensor signal, which shows a ripple, due to the exposure to a visible or UV light modulation, on top of the resistance changes related to the sensor interaction with gases. During the semi-period where the modulating light is off, just the reaction of the sensing material with the target gas takes place at the sensor surface, while in the semi-period where the light is on, photons from the light create electron-hole pairs which participate in the current conduction, and in addition, they promote the desorption of surface adsorbed species, thus, modulating the resistance from the sensing material. Tungsten trioxide is an n-type MOX, which has a bandgap of about 2.7 eV [58]. Strontium titanate is an n-type perovskite oxide with a bandgap of about 3.2 eV [59]. Tungsten disulphide is a p-type semiconductor that has a bandgap of about 1.3 eV [60,61]. The photon energies of the used purple (visible) and UV LEDs are 3.02 eV and 3.40 eV, respectively. Regarding the SrTiO_3_@WO_3_ sensor, an equilibrium between the Fermi levels takes place due to the formation of a heterojunction at the interface. Thus, even though the creation of electron-hole pairs is not promoted in the SrTiO_3_ by visible light due to the photon energy be lower than its bandgap, it acts as a catalyst to promote the separation of the electron-hole pairs, which provides redox reaction sites [59].

#### Models based on FFT Components

Based on the results obtained in [56] the pulsed light ON/OFF period was set to 20 s to have a higher number of light pulses within the analyzed time. In order to shorten the number of samples used in the analysis, a period of 2 min from the time domain signal was selected to perform the FFT (shown in Figure 2a). Thus, having a sampling rate of 1 Hz, vectors used to perform the FFT have 120 values, which is equivalent to six light ON/OFF pulses. Due to the time needed to establish a homogeneous gas concentration inside the chamber, the first 6 min of each gas pulse is not used in the analysis. Hence, the 7th and 8th minutes (counted from the gas cycle start) of each gas pulse are used to create the vectors employed to perform de FFT analysis. After carrying out the FFT analysis, vector size is halved. Hence, the frequency components vector obtained (related to each gas cycle) has half of the size with respect to the time domain signal vector. However, not all the frequency components are used to build the training matrix used to develop the PCR calibration models. FFT vectors are manipulated to use just the components which give relevant information from the sensor signal. As Figure 2b shows, the switching frequency of the pulsed light (0.05 Hz for an ON/OFF period of 20 s) and its even order harmonics appear in the frequency spectrum. Hence, to reduce the number of components used to build the training matrix and eliminate low-intensity frequency components, which can be affected by noise, just frequency components with a relevant magnitude are taken. The training matrix is built by concatenating the new vectors related to each concentration. Rows (observations) represent different concentrations, and columns (variables) are each of the frequency components used. The training matrix built with the frequency components is used to perform the PCA and the principal components (PC) obtained in this process are used to perform the PCR calibration models. In addition, it is possible to build the training matrix using frequency components related to more than one sensor or including observations from different gases. Hence, the scores and loadings plots obtained from the PCA are useful to identify different gases and distinguish the contribution of each PC to the discrimination performance. The accuracy of both n-type (WO_3_ and SrTiO_3_@WO_3_) and p-type (WS_2_) sensors to quantify oxidizing (NO_2_) and reducing (NH_3_) gases was tested by performing calibration models and these were cross-validated, with the combined use of low operating temperature (50 °C) and light modulation. In addition, the effect of applying light modulation, exciting the sensor surface with LEDs having wavelengths in the ultraviolet and visible spectrum was evaluated. Results from the prediction model accuracies are evaluated through the R-squared (R^2^) and Root Mean Square Error (RMSE) values.

Cross-validation methods are used to do the validation process and evaluate the model’s accuracy to predict the target gas concentration. Hence, a leave-one-out strategy is applied cyclically. First, the data related to one of the cycles of 3 concentrations is left out of the training matrix, and the PCR is performed with the rest of the data. Then, the beta values obtained from the PCR are used with the new data (left out data) to identify the gas concentration and validate the methods. Once the strategy is applied to all the data, the validation model is obtained by concatenating each set of data identified. 

## 3. Results and Discussion

### 3.1. Morphological Characterization

Figure 3 depicts typical FESEM images from pure WO_3_ NNs at 2 different magnification values. The EDX spectrum (see Appendix A) shows that the WO_3_ NNs are composed of tungsten and oxygen, being the sample free of any contaminant. Raman spectroscopy was also employed. From the Raman spectrum (shown in Figure 4), the position and intensity of the bands at 807, 717, 325, and 274 cm^−1^ are typical from the monoclinic phase of WO_3_ [62,63,64]. XRD analysis results (see Appendix A), also confirm the presence of the monoclinic phase of WO_3_.

Figure 5 shows typical FESEM images where the morphology of the nanoneedles for strontium titanate loaded WO_3_ is revealed. The tips of loaded NWs present a granular morphology. EDX analysis does not show the presence of Sr or Ti in loaded samples. In fact, the EDX spectrum for strontium titanate loaded WO_3_ is identical to the one shown in Appendix A for the pristine WO_3_ material, so it can be concluded that the granular morphology at the tips of NWs corresponds also to WO_3_. Raman spectroscopy and XRD were also performed and neither the spectrogram nor the diffractogram show peaks that indicate the presence of SrTiO_3_. After these analyses, it was possible only to confirm the presence of WO_3_ in its monoclinic phase. XPS and ToF-SIMS were also used to evaluate the surface composition. The XPS spectrum recorded on the SrTiO_3_@WO_3_ sample is shown in Appendix A, the peaks generated by photoelectrons emitted from W, O, and C atoms are clearly recognized. The relative amount of each observed element was O 24% at., W 70% at., and C 6% at., the detailed analysis of the W 4f indicates that the oxidation state of the W atoms is +6. XPS did not detect the presence of SrTiO_3_ at the sample surface. Finally, ToF-SIMS was considered for investigating the presence of strontium titanate in loaded WO_3_ samples, due to the higher sensitivity of this technique to detect trace elements in comparison to any of the previously used ones. The ToF-SIMS spectra (see Appendix A) confirm the presence of Sr and Ti. It is therefore concluded that loaded samples contain strontium titanate, on the surface of WO_3_ but at low concentrations (i.e., below the detection threshold of XRD and XPS).

The as-grown nanofilms of WS_2_ were strongly adherent to the substrate with dark black color. The results obtained revealed that the WO_3_ NNs morphology changed completely to form nanoflakes of WS_2_, which can be well-identified in Figure 6. Furthermore, it can be seen that these nanoflakes are assembled in a 3D topology and appear as nanoflowers.

From the EDX spectrum shown in Appendix A, it is confirmed that the composition of the as-grown nanoflakes of WS_2_ consists of sulfur and tungsten. No oxygen peak is identified in the EDX spectrum, which apparently confirms the development of a high-yield WS_2_ phase, free from oxide content. Also, the grown material was characterized using Raman spectroscopy to confirm its purity. From the Raman spectrum (shown in Figure 7), 2 important Raman peaks, characteristic of 2H-WS_2_ were observed at 348 and 414 cm^−1^. Additionally, two broad peaks with very low intensity were also detected at 701 and 804 cm^−1^, indicating the presence of some WO_3_ impurities that could be present in the bulk of the grown material [65]. 

XPS was used to evaluate the formation of W-S bonds. The peaks shown in Appendix A, corresponding to the S 2p_1/2_ and S 2p_3/2_ orbital of divalent sulfide ions, are observed at 163.3 and 162.1 eV. The W peaks shown in Appendix A located at 38.3, 34.7, and 32.5 eV correspond to W 5p_3/2_, W 4f_5/2_, and W 4f_7/2_, respectively. The energy positions of these peaks indicate a W valence of +4, which is in accordance with the previous reports. The other doublet with components at W 4f_5/2_, and W 4f_7/2_ respectively at 30.8 and 35.8 eV indicates the presence of W-O in WO_3_. The Raman spectroscopy, EDX, and XRD results (Appendix A) did not indicate the presence of tungsten oxide. As XPS is sensitive to the near surface region, while the other techniques probe much deeper below the surface, the comparison of the results of these different techniques indicate that the oxide is mainly located near the surface.

In summary, the sulfurization process conducted on WO_3_ NNs yields a 3D assembly of WS_2_ nanoflakes with a small amount of WO_3_ impurities, as revealed by Raman and XPS.

### 3.2. Gas Sensing Characterization

#### 3.2.1. Standard Operation

After the morphological and compositional characterization of the synthesized materials, sensors were tested for gas sensing. In the first stage, the sensors were activated by heating their active films and without light modulation. Using the procedures described before, the gas sensing properties were investigated at the operating temperatures of 50, 100, and 150 °C. When the n-type sensors (WO_3_ and SrTiO_3_@WO_3_) were exposed to an oxidizing gas (NO_2_) and the p-type sensor (WS_2_) to a reducing species (NH_3_), their response monotonically increased as the temperature was raised. Thus, the highest responses were obtained when the operating temperature was set at 150 °C. Appendix A summarizes these results. In contrast, n-type sensors presented a poor response reproducibility towards NH_3_ for all the operating temperatures tested. Similarly, the p-type sensor presented also reproducibility issues when exposed to NO_2_. In conclusion, for the range of operating temperatures studied, WO_3_ and SrTiO_3_@WO_3_ sensors are more suited for detecting nitrogen dioxide, while WS_2_ is more suited for detecting ammonia.

#### 3.2.2. Pulsed Light Modulation

The input of each frequency component to the gas identification process was evaluated through biplots performed with the scores and loadings from the PCA. The analysis of the frequency components selection for performing the target gas identification is presented in Appendix A. Results from this analysis show that using just the ON/OFF frequency related to the SrTiO_3_@WO_3_ sensor and its first even order harmonic contains enough information for discriminating between the two species considered. This is true for visible and UV light excitation. Furthermore, using these two frequency components it is possible to separate observations related to different NO_2_ concentrations in the scores plot. In the same way, frequency components (ON/OFF and its first even order harmonic) extracted from the WS_2_ sensor allow separating NH_3_ concentrations in different groups in the scores plot. Thus, all the training matrices used to obtain the PCA scores plots presented in Figure 8 were built using just the light switching frequency and its first even order harmonic. If the WO_3_ pristine sensor is used to perform the same analysis, the PCA scores plot allows to discriminate between NH_3_ and NO_2_ observations, but when the sensor works under UV light modulation the ability to separate each gas concentration worsens.

Figure 8a,b show the PCA scores plot obtained from a training matrix built using observations from both NO_2_ and NH_3_ and frequency components related to the SrTiO_3_@WO_3_ sensor, under visible and UV light modulation, respectively. It is clear that NH_3_ and NO_2_ observations can be separated into different clusters according to the PC1. In a supposed real application, this would allow the identification of the target gas for using the proper model to quantify the gas concentration. Different NO_2_ concentrations can be also identified in clusters separated according to PC1. 

Figure 8c,d depict the PCA scores plots obtained when the training matrix is made using just NO_2_ observations and frequency components from the SrTiO_3_@WO_3_ sensor, working under visible and UV light modulation, respectively. In both cases, the different gas concentrations can be grouped and separated according to the PC1, which allows to perform a qualitative identification of the concentration. Figure 8e and f show the PCA scores plot obtained from a training matrix built with just observations of NH_3_ and frequency components from the WS_2_ sensor when it works under visible and UV light modulation, respectively. In this case, when the WS_2_ sensor works under visible light it is possible to identify different clusters for each concentration organized according to the PC1, although higher concentration clusters are close together. When the WS_2_ sensor is operated under UV light modulation, the clusters corresponding to different concentrations can be separated as well, although some of the 50 and 75 ppm observations are overlapped. In this case, the cluster orientation is diagonal due to a different distribution of the variance explained by each principal component with respect to when the sensor is operated under visible light modulation. 

From these results, it is deduced that under light pulse modulation n-type sensors are useful for quantifying oxidizing species (NO_2_) and p-type sensors are suitable for quantifying reducing species (NH_3_). PCR models built for predicting concentration are discussed below.

According to the results obtained with the principal component analysis, the PCR calibration models, and cross-validation results presented in Figure 9 and Figure 10 were obtained using scores and loadings data from 1st and 2nd principal components obtained from the PCA developed using just two frequency components (light switching frequency and its first even order harmonic). These two PCs explain over 99% of the data variance. PCR models related to the WO_3_ pristine sensors are presented in Appendix A.

Figure 9a,b illustrates the WS_2_ and SrTiO_3_@WO_3_ sensors results for NH_3_ and NO_2_, respectively, under UV light modulation. WS_2_ sensor model presents an R^2^ value of about 0.90 and its RMSE value is about 13% for the total measured concentration range. On the other hand, the SrTiO_3_@WO_3_ sensor model presents an R^2^ value near 0.97 and its RMSE value represents just 7.44% of the total measured concentration variation. Results obtained make the models suitable for quantifying and predicting the target gas concentrations.

Figure 10 shows how, for the two sensor types, R^2^ values are higher than 0.92 and RMSE values represent near or less than 10% for the concentration measured range. Results obtained when the sensors are working under visible light modulation are better than those when UV light is used. The model obtained with the SrTiO_3_@WO_3_ sensor reaches almost 0.98 of R^2^ and has an RMSE value that represents just about 5% of the total measured concentration range. The performance of all sensors when quantifying gas concentrations is better under visible light modulation than under UV modulation. From the results obtained it may be deduced that the SrTiO_3_ loading gives more stability to sensor response and makes this hybrid more suitable for being used to predict NO_2_ concentrations than using pure WO_3_ sensors.

To further support these conclusions, PCR models were also built and validated for n-type sensors to quantify ammonia and for p-type sensors to quantify nitrogen dioxide (see Appendix A) achieving bad performance, as foreseen.

Moreover, the system performance towards gas mixtures was tested. The tests consisted of keeping the NH_3_ concentration fixed while variating the NO_2_ concentration (250, 500, and 750 ppb) using the gas cycles and baseline recovery time exposed in Section 2.3. This set was repeated twice, working at two different NH_3_ concentrations (4 and 15 ppm). The PCA scores from the results of these tests allow discrimination between observations when the presence of single gases is detected (just NO_2_, just NH_3_) and when a mixture of these gases is present. PCR models developed for single gas concentration quantification were not accurate to quantify gas mixtures. Further study is needed to quantify the gas concentration in presence of gas mixtures using multivariate analysis methods.

The methods implemented here suppose a reduction in the time needed to identify the gas concentration in comparison to the process presented by Gonzalez et al. in [31], and even an improvement of the results obtained in [56]. Also, the combined use of low temperature and light modulation allows a power consumption reduction of about 90% as compared to the traditional thermal activation of MOX sensors (using the same substrates as in the present work), where operating temperatures of 100 – 500 °C are used. Moreover, sensors were operated under pulsed light modulation for over one month and the morphology of nanomaterials remained unchanged and so was their response to the species tested. In addition, the cost of visible-light LEDs is 10 times lower than that of UV light LEDs used here, and 250 times lower than the cost of the UV light LEDs used in [56].

## 4. Conclusions

In this paper, WO_3_ and SrTiO_3_@WO_3_ (n-type) and WS_2_ (p-type) sensors were synthesized and characterized. The combination of UV or visible pulsed light modulation with low temperature was employed to modulate the resistance of sensors in a background of oxidizing or reducing species. The use of pulsed light modulation, FFT analysis, PCA, and linear regression techniques for building predictive models to identify and quantify gases has been implemented. PCA scores enable the discrimination between the two different target gases (NO_2_ and NH_3_). Prediction models with up to 0.98 of R-square value and RMSE value lower than 10% over the total concentration range measured were obtained. The sensing layer activation mechanism applied enables a power consumption reduction of more than 90% in comparison to the one of traditional high temperature operated MOX non-MEMS sensors. Moreover, the sensor signal period used to quantify target gases was reduced with respect to previously published results, thus shortening the time needed for quantification. Using visible light (410 nm) led to better results than using UV light (365 nm). On the other hand, it was demonstrated that p-type sensors achieved better performance to quantify reducing gases, while it was confirmed that n-type sensors exhibit higher efficacy to quantify oxidizing gases. The loading of SrTiO_3_ nanoparticles to WO_3_ led to better results both in the discrimination between gases and the quantification of oxidizing species. The methodology presented in this work opens an opportunity to use non-MEMS MOX sensors in real gas sensing applications, since reducing and oxidizing gas concentrations can be accurately quantified using a short period of sensor signal and thus, saving a considerable amount of power.

## Figures and Tables

**Figure 1 sensors-21-03736-f001:**
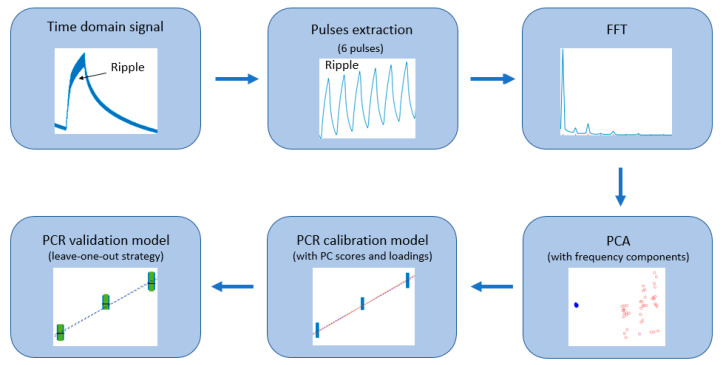
Data analysis process flow diagram.

**Figure 2 sensors-21-03736-f002:**
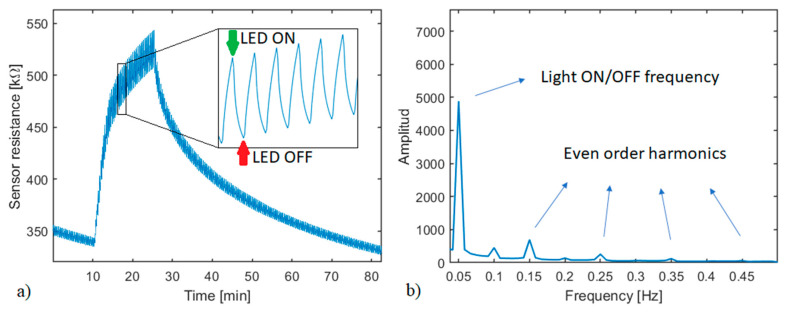
(**a**) Time-domain data extraction from the sensor response to a gas pulse to perform the FFT, 2 min (6 light pulses) of signal are used. (**b**) Frequency domain obtained after applying the FFT. The light switching frequency and its even order harmonics are specified.

**Figure 3 sensors-21-03736-f003:**
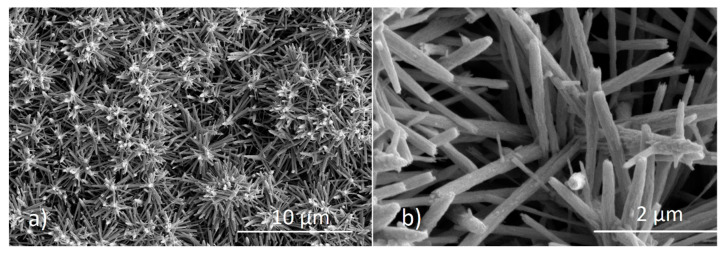
Tungsten trioxide nanoneedles, (**a**) 6500× of magnification and (**b**) 37,400× of magnification. Working with an electron beam acceleration voltage of 5 kV.

**Figure 4 sensors-21-03736-f004:**
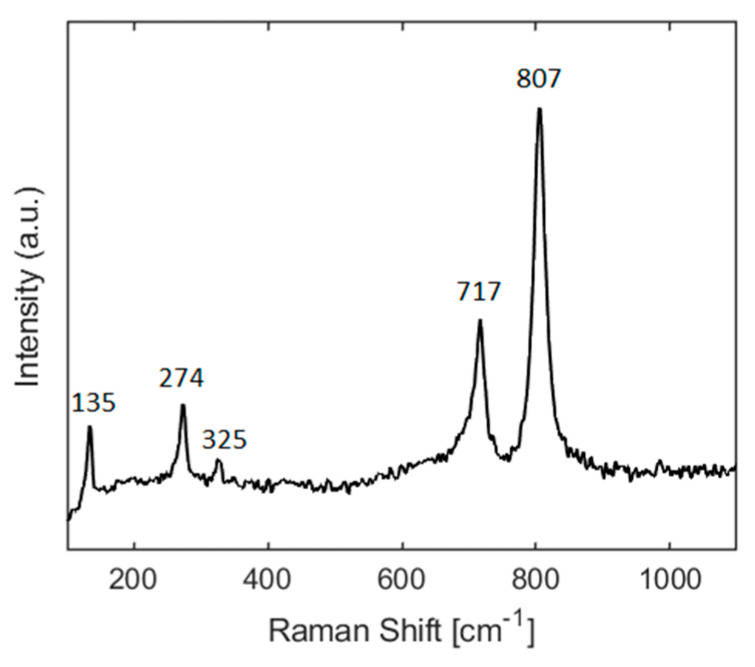
Raman spectrum from the WO_3_ sensor measured using 633 nm excitation.

**Figure 5 sensors-21-03736-f005:**
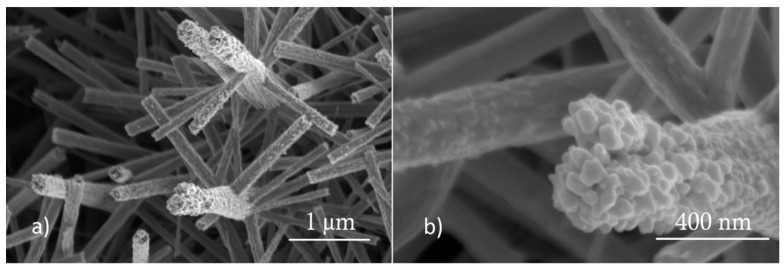
SrTiO_3_@WO_3_ sensor surface, (**a**) 41,000× magnification with an electron beam acceleration voltage of 2 kV, and (**b**) 150,000× magnification and an electron beam acceleration voltage of 5 kV.

**Figure 6 sensors-21-03736-f006:**
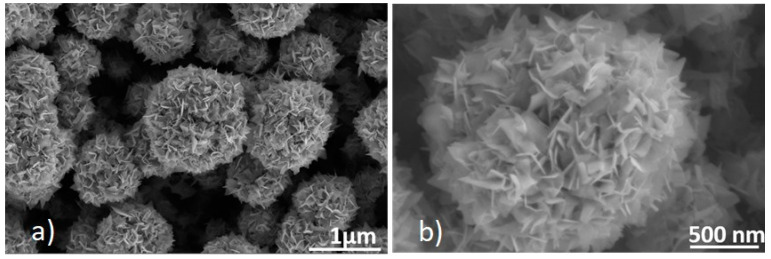
FESEM image depicting nanoflakes of WS_2_ assembled to form nanoflowers. Magnification of (**a**) 25,000× and (**b**) 65,000×. The electron beam acceleration voltage was set at 5 kV.

**Figure 7 sensors-21-03736-f007:**
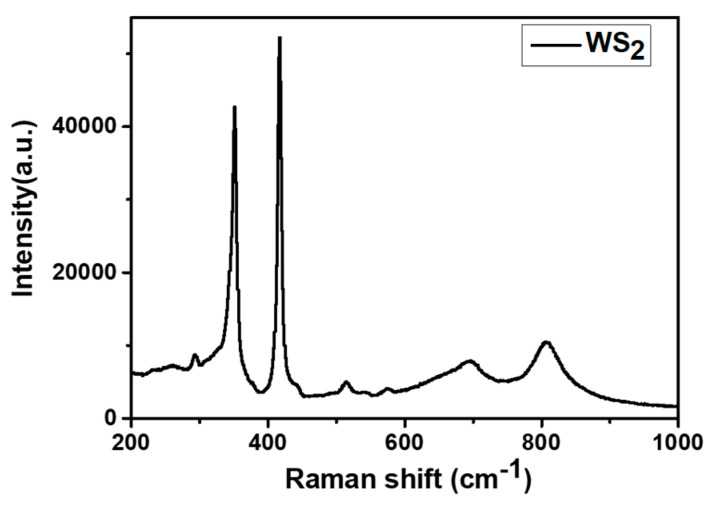
Raman spectrum from the WS_2_ sensor.

**Figure 8 sensors-21-03736-f008:**
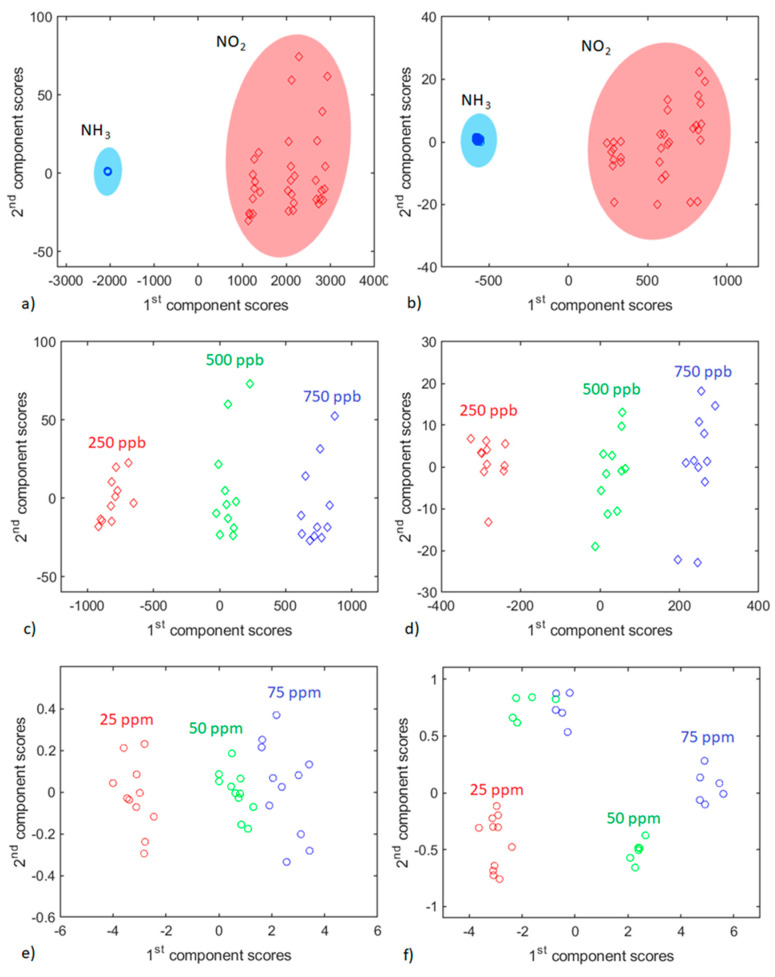
PCA scores plot for (**a**,**c**,**e**) visible light modulation and (**b**,**d**,**f**) UV light modulation using two frequency components from WS_2_ or SrTiO_3_@WO_3_ sensors. PCA from a and b were performed using observations of NO_2_ and NH_3_ to construct the training matrix and frequency components from the SrTiO_3_@WO_3_ sensor, while c, d, and e, f belong to PCA developed with observations of just NO_2_ and just NH_3_, and frequency components from the SrTiO_3_@WO_3_ sensor and the WS_2_ sensor, respectively. In the figure, circles represent NH_3_ observations and diamonds represent NO_2_ observations. The first two PCs explain over 99% of data variance.

**Figure 9 sensors-21-03736-f009:**
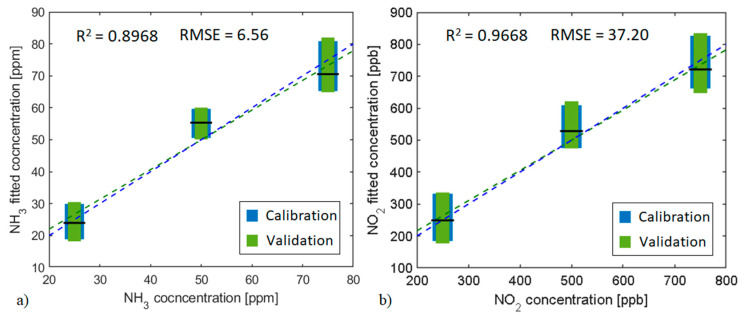
PCR calibration model and cross-validation for the (**a**) WS_2_ sensor towards NH_3_ concentrations and (**b**) SrTiO_3_@WO_3_ sensor towards NO_2_ concentrations. The operating temperature was 50 °C and the light modulation was done with UV LEDs. Blue boxes represent the calibration model dispersion for each concentration and green boxes the validation dispersion. The horizontal black line represents the mean value for the validation process. The validation linear fit is shown with the green dashed line, and the blue dashed line represents a unitary slope line. R-squared and RMSE values belong to the calibration model.

**Figure 10 sensors-21-03736-f010:**
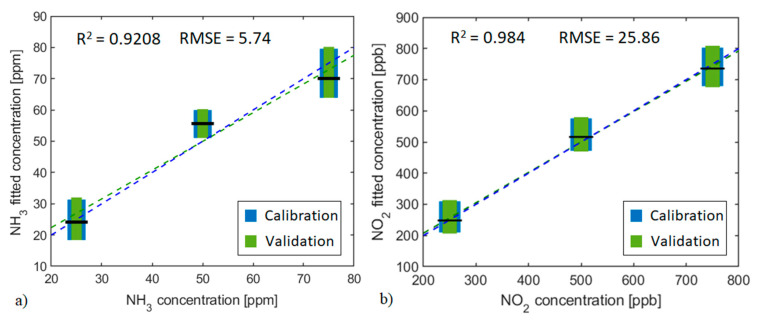
PCR calibration model and cross-validation for the (**a**) WS_2_ sensor towards NH_3_ concentrations and (**b**) SrTiO_3_@WO_3_ sensor towards NO_2_ concentrations. The operating temperature was 50 °C and the light modulation was done with purple visible light LEDs. Blue boxes represent the calibration model dispersion for each concentration and green boxes the validation dispersion. The horizontal black line represents the mean value for the validation process. The validation linear fit is shown with the green dashed line, and the blue dashed line represents a unitary slope line. R-squared and RMSE values belong to the calibration model.

## Data Availability

Raw data are available from authors upon request.

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
