# Peer review of "On the Use of Pulsed UV or Visible Light Activated Gas Sensing of Reducing and Oxidising Species with WO3 and WS2 Nanomaterials"

_sensors, 2021, doi:10.3390/s21113736_

Round 1

Reviewer 1 Report

What has been presented in this paper seems to create a platform to address one of the existing  challenges in metal oxide gas sensors, however I would like to ask the authors to include below minor changes in the final version:

1- Abstract can be rewritten to reflect more of what have been presented in conclusion.

2- Line 43: Please give a few examples for “A wide range of different applications” since MOX has also limitations due to their high operating temperature.

3- Experimental Set-up: Please add a few figures of experimental setup to make it more clear.

4- Line 484: Please revise 3 to lowercase in SrTiO3.

Author Response

See the answers in the attached file.

Reviewer 2 Report

This paper presented a novel way to modulate MOX sensors which in combination with post processing, can increase the sensor selectivity to discriminate between target gases. The results shown were promising, so I would recommend this paper for publication with some minor changes.

  1. In the experiment setup, the gas testing setup was described. It will be better to include a photograph of the test setup, or a block diagram together with the explanation. The diagram could also include the pulsed light setup.
  2. In the result section, the explanation for the figures was a bit difficult to follow. For instance, figure 9 was explained from line 382 to 388, but then again from line 440 to 446. Although there were from different aspects of the plot, it will still be easier to read if the explanations for the figures are combined and in the order of their appearances. In addition, line 419 mentioned figure 11 which is not included in the paper, maybe this is a typo?
  3. On page 13, it was mentioned that the light modulation method consumes less power. So in what range is the power consumption? Few mW or less? A rough estimation is preferred.
  4. Some minor typos and formatting issues: in conclusions, it is MEMS instead of MEMs; and the format for reference 1 is different from the rest.

Author Response

See the answers in the attached file.

Reviewer 3 Report

The authors present the W)3 and WS2 based gas sensors using UV and visible light activation responses. The authors claimed an improved response for reducing and oxidizing gas species using light modulations. The manuscript's presentation value is high and the data is fairly organized. However, I don't see an novelty in the presented results. The idea is so obsolete that it has already been exploited for a number of times for a long time in research. Hence, I would like to decide based on the major revision of the following comments.

1- The author are unclear as to either they want to use UV light or Visible light or both? 

2- Line 44: The gas molecules react with the oxygen vacancies or the adsorbed ambient oxygen?

3- Provide the CAS numbers of all the chemicals and substrates used in the experiments.

4- Section 2.3: Provide a schematic figure explaining all the major and minor components of the gas measuring setup. Also, provide the real-time digital image of the setup for clarity.

5- Did the authors exposed the system for both gases together to test the selectivity of the sensors?

6- Figure 3 and figure 6: Which figure is a and which one is b?

7- Did the authors witnessed any change in response based on the change in nanomaterials' morphology or dimensions?

8- In the introduction section, the authors argued that high sensing temperature is one of the biggest drawbacks of MOX sensors and the industry needs replacement. However, the working temperature of sensors, without light illumination, in this work is also way above the room temperature. Then, how would the presented sensors be different from the time tested and reliable MOX sensors? 

9- There are plenty of studies already available who have studied the gas sensors' response using light modulations. Although, I don't agree with the proposed mechanism of the light modulation techniques; however, the biggest question lies here is that what is the novelty of this study provided the technique has already been exploited a lot for gas sensors? It gives me an impression that the study doesn't add value to science but just presented for the sake of publication. Elaborate in detail.

10- Where is the supporting information?

Author Response

See the answers in the attached file.

Round 2

Reviewer 3 Report

The authors have provided a comprehensive response to the queries raised in review round 1. I believe the manuscript has been improved a lot and merits publication in sensors. Hence, I would like to accept paper publication in Sensors in the present form.